# Differentiating axonal loss and demyelination in chronic MS lesions: A novel approach using single streamline diffusivity analysis

Samuel Klistorner[1], Michael H. Barnett[2,3], Jakob Wasserthal[4], Con Yiannikas[5], Joshua Barton[2], John Parratt[5], Yuyi You[1,6], Stuart L. Graham[6], Alexander Klistorner[1,6]*

1 Save Sight Institute, Sydney Medical School, University of Sydney, Sydney, Australia, 2 Brain and Mind Centre, University of Sydney, Sydney, New South Wales, Australia, 3 Sydney Neuroimaging Analysis Centre, Camperdown, New South Wales, Australia, 4 Division of Medical Image Computing (MIC), German Cancer Research Center, Heidelberg, Germany, 5 Royal North Shore Hospital, Sydney, New South Wales, Australia, 6 Faculty of Medicine and Health Sciences, Macquarie University, Sydney, New South Wales, Australia

* sasha@eye.usyd.edu.au

**Data Availability Statement:** There are ethical restrictions on sharing the data set, which contains potentially sensitive patient information. However,

## Abstract

We describe a new single-streamline based approach to analyse diffusivity within chronic MS lesions. We used the proposed method to examine diffusivity profiles in 30 patients with relapsing multiple sclerosis and observed a significant increase of both RD and AD within the lesion core (0.38+/-0.09 $\mu m^2$/ms and 0.30+/-0.12 $\mu m^2$/ms respectively, p<0.0001 for both) that gradually and symmetrically diminished away from the lesion. T1-hypointensity derived axonal loss correlated highly with ΔAD (r = 0.82, p<0.0001), but moderately with ΔRD (r = 0.60, p<0.0001). Furthermore, the trendline of the ΔAD vs axonal loss intersected both axes at zero indicating close agreement between two measures in assessing the degree of axonal loss. Conversely, the trendline of the ΔRD function demonstrated a high positive value at the zero level of axonal loss, suggesting that even lesions with preserved axonal content exhibit a significant increase of RD. There was also a significant negative correlation between the level of preferential RD increase (ΔRD-ΔAD) in the lesion core and the degree of axonal damage (r = -0.62, p<0.001), indicating that ΔRD dominates in cases with milder axonal loss. Modelling diffusivity changes in the core of chronic MS lesions based on the direct proportionality of ΔAD with axonal loss and the proposed dual nature of ΔRD yielded results that were strikingly similar to the experimental data. Evaluation of lesions in a sizable cohort of MS patients using the proposed method supports the use of ΔAD as a marker of axonal loss; and the notion that demyelination and axonal loss independently contribute to the increase of RD in chronic MS lesions. The work highlights the importance of selecting appropriate patient cohorts for clinical trials of pro-remyelinating and neuroprotective therapeutics.

## Introduction

Diffusion tensor imaging (DTI) is sensitive to the microstructural organisation of white matter tracts and has been suggested as a new promising tool that provides greater pathological specificity than conventional MRI, helping, therefore, to elucidate disease pathogenesis and

other researchers may send data access requests to the Macquarie University Human Ethics Committee (human.ethics@mq.edu.au).

**Funding:** AK-National Multiple Sclerosis Society (NMSS), Novartis Save Neuron Grant, Sydney Eye Hospital foundation grant and Sydney University Medical foundation. The funders had no role in study design, data collection and analysis, decision to publish, or preparation of the manuscript.

**Competing interests:** Study was partially funded by Novartis. This funds was used exclusively for MRI data acquisition. None of the authors were employed or recieved consultancy from commercial entity or participated in patents, products development or marketing. This does not alter our adherence to PLOS ONE policies on sharing data and materials.

monitor therapeutic efficacy. While axial diffusivity (AD) has been linked to axonal pathology, radial diffusivity (RD) has been suggested as a surrogate biomarker associated with the level of white matter myelination [1–3]. This is of particular importance since recent interest in the development of remyelinating therapies [4–6] has increased demand for reliable and standardized techniques capable of assessing remyelination *in vivo*.

The main advantages of the DTI technique are its simple single-shell protocol, relatively fast data acquisition and straightforward analysis. While more complex models [7–9] may help to better explain the observed experimental or clinical data [10], they require very high quality acquisition and measurements that may not be feasible in a clinical setting. Therefore, DTI remains the method of choice in clinical studies/trials, particularly when multiple sites are involved.

However, apart from intrinsic white matter properties, the alterations in DTI metrics can also be affected by the degree of fibre coherence. The majority of white matter voxels are characterized by extensive crossing, kissing, bending or fanning of fibers [11]. Therefore, apart from the limited area of white matter occupied by coherent fiber tracts, voxel-based analysis utilising DTI models does not adequately estimate orientationally-sensitive diffusivity measures, such as axial and radial diffusivity (AD and RD) [12, 13].

In order to isolate disease-related DTI changes from alterations related to fiber coherence the comparison of pathological tissue must be made with an area of the brain which displays similar fiber structure. Thus, attempts have been made to use symmetrical white matter areas of the opposite hemisphere to control for fiber non-homogeneity. This process, however, is challenging and require manual input. In addition, MS lesions frequently affect symmetrical (periventricular) parts of the brain.

An alternative approach utilising an analysis of fiber tract diffusivity profiles has recently been suggested [14–16]. Using this technique, profiles of various diffusivity measures (such as fractional anisotropy, mean, radial and axial diffusivity) of an individual coherent white matter fiber tract in a subject or group of subjects are constructed between the two regions of interest (ROIs) and compared to a similar fiber tract in a different group (for example, non-MS controls).

This approach has recently been extended by segmenting a single coherent tract into areas bound by seemingly similar pathological processes (such as MS lesions), which allows the separation of lesional fibers (i.e. fibers crossing the lesion) from non-lesional fibers of the same tract. Due to similar coherency of corresponding lesional and non-lesional fibers this technique facilitates within-subject comparison of diffusivity measures between normal and pathological tissue by providing "internal" reference [17].

However, MS lesions are rarely confined within a single (coherent) fiber tract. Rather, multiple crossing tracts frequently traverse individual lesions (or part thereof) in different directions; the presence of crossing fibers, therefore, similarly limits existing implementations of the profile-based technique. In addition, due to the irregular shape of MS lesions, the intralesional length of individual fibers can vary considerably, degrading the accuracy of diffusivity measurements inside and outside of lesions.

In an effort to expand this approach beyond the coherent fiber tracts we developed a new fully automated single-fiber based technique to analyse diffusivity alteration within MS lesions and its close surrounding. The method is based on computation of the difference (asymmetry) between the diffusivity profile of an individual streamline passing through the lesion of interest ("lesional fiber") and the average diffusivity profiles of several adjacent and similarly oriented non-lesional streamlines of equal length and orientation, obtained from the same fiber tract ("non-lesional fibers"). Similar to previously described technique used in coherent fiber tracts, this method provides "internal" reference for diffusivity measure within MS lesions. In

addition, this technique eliminates the ambiguity of the fiber-based analysis caused by irregular lesion shape. Furthermore, based on based on an observation that distribution of mean diffusivity in the brain white matter is relatively uniform (see S1 Fig), we developed an algorithm to minimise the residual effect of fiber non-coherency between corresponding voxels of lesional and non-lesional streamlines.

Using this technique, we estimated asymmetry profiles of axial and radial diffusivity ($\Delta$AD and $\Delta$RD) in the core and rim of individual lesions and in surrounding normal appearing white matter; and computed the personalised (subject-based) diffusivity profile in 30 patients with relapsing remitting MS (RRMS). In addition, we estimated the relative contribution of demyelination and axonal damage to the elevation of RD in chronic MS lesions.

## Method

### Standard protocol approvals, registrations, and patient consents

The study was approved by University of Sydney and Macquarie University Human Research Ethics Committees and followed the tenets of the Declaration of Helsinki. Written informed consent was obtained from all participants.

### Subjects

Thirty consecutive patients with RRMS, defined according to the revised McDonald 2010 criteria, were enrolled [18].

### MRI protocol

The following sequences were acquired using a 3T GE Discovery MR750 scanner (GE Medical Systems, Milwaukee, WI):

1. Pre- and post-contrast (gadolinium) Sagittal 3D T1

2. FLAIR CUBE

3. diffusion weighted MRI

   Specific parameters are presented in S1 File.

### MRI image pre-processing

The baseline T1-weighted imaging was realigned to AC-PC orientation in MrVista package (Stanford University). Diffusion MRI was corrected for motion, eddy-current distortion in FSL and EPI susceptibility distortion using blip-up/ blip-down sequences.

Subsequently, tensor reconstruction was performed in MrDiffusion (MrVista, Stanford University). Tensor images were then linearly co-registered to corresponding T1 AC-PC images.

### Lesion identification and analysis

Individual lesions were identified on the co-registered T2 FLAIR images and semi-automatically segmented using JIM 7 software (Xinapse Systems, Essex, UK) by a trained analyst.

The core of the lesion was identified by shrinking the lesion mask in all directions by 1 voxel using the "eroding" function of JIM software. The rest of the lesion was assigned as the "rim" area. Only lesions measuring larger than 100 mm$^3$ (i.e.100 voxels on T2 FLAIR image) were selected for analysis. Gadolinum (Gd)-enhancing lesions (detected in 2 patients) were excluded from the analysis.

The lesion mask was also applied to pre-contrast 3D-T1-weighted images to quantify lesion hypointensity. Since some lesional voxels (particularly in lesions with a high degree of tissue destruction) did not contain any traversing fibers, only voxels which intersected with lesional fibers (see below) were selected for T1-hypointensity calculation.

In order to reduce inter-subject variability, lesional T1-hypointensity was normalised by the intensity of NAWM, which was measured using two additional 5 mm ROIs placed in the NAWM of both hemispheres. In addition, the minimum intensity of CSF on T1-weighted images was measured by placing 2 mm ROIs inside anterior horns of lateral ventricles.

## Identification of major fiber tracts

TractSeg algorithm was used to identify 72 anatomically well-defined tracts as described by Wasserthal and co-workers [19, 20]. TractSeg uses a fully convolutional neural network to directly segment these tracts, taking the fiber orientation distribution function (fODF) peaks as input. To generate tract-specific tractograms, TractSeg also generates so called tract orientation maps (TOMs) that can be used together with the tract segmentations to generate high-quality tractograms of 72 major tracts. For each tract, 2000 streamlines were generated.

TractSeg only generates binary tract masks, which, however, make it easy to generate streamlines since seeding for probabilistic tractography is produces within the mask and certain number of streamlines are generated for each tract. The TractSeg methodology through MS lesion is described in [20].

## Single fiber-based lesional profile

Each individual lesion was intersected with all 72 tracts and tracts overlapping with lesions were selected for analysis (Fig 1).

Each lesional streamline was analysed separately.

Firstly, the lesional streamline ("lesional fiber") was limited to five mm on each side of the lesion and limiting ROIs perpendicular to lesion streamline were constructed. Then, a minimum of 5 adjacent streamlines not overlapping with any lesion ("non-lesional fibers"), but of similar length and orientation to the lesional fiber, were selected from the same fiber tract to be used as a "local reference" (Fig 2a) (lesional fibers which did not have corresponding non-

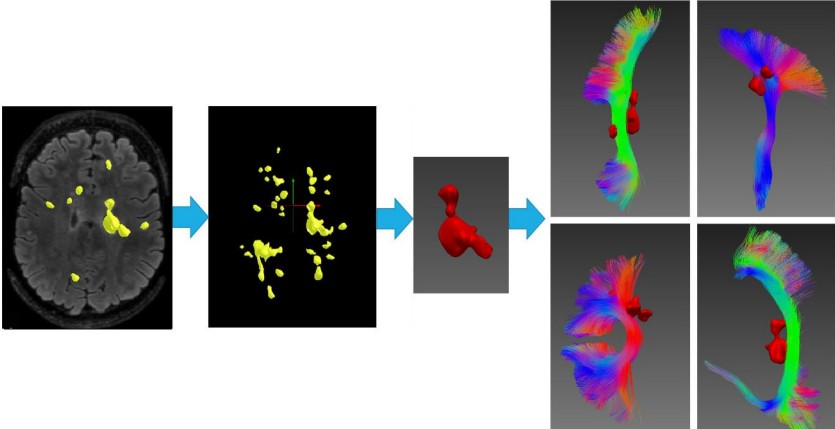

**Fig 1. Selection of fiber tracts.** Lesions (yellow) identified on FLAIR images (2D slice and 3D lesion mask). Individual lesion mask (red) intersected with each of the 72 fiber tracts. Examples of 4 fiber tracts intersected with a single lesion are presented.

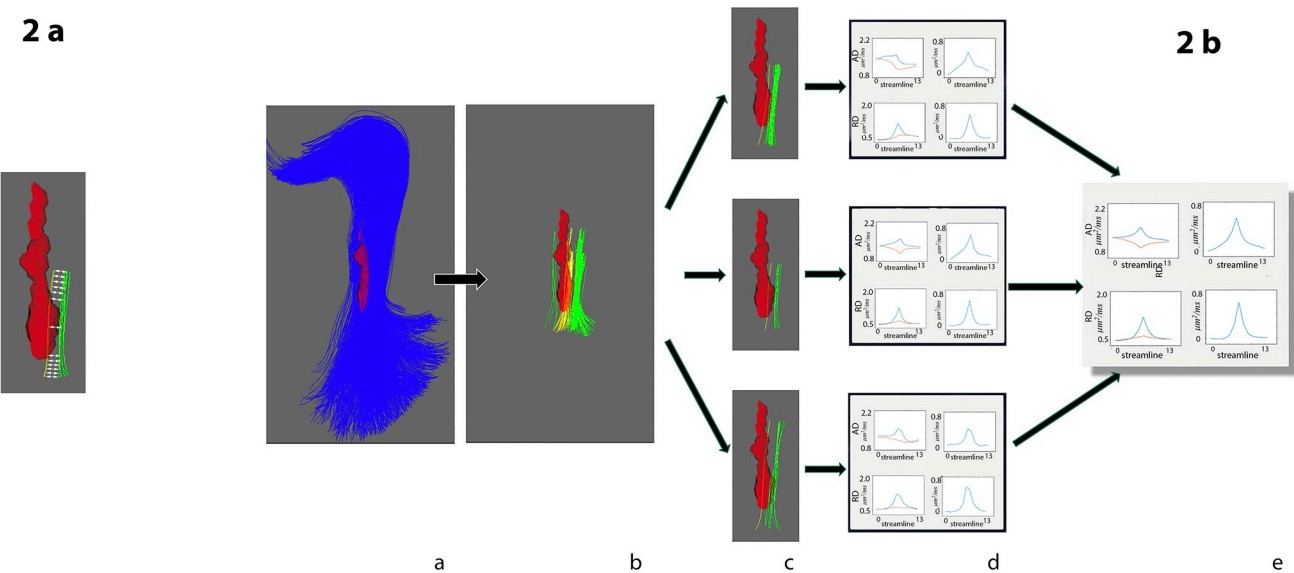

**Fig 2. Constructing single fiber tract diffusivity profile of individual lesion.** a) Corresponding points of lesional (yellow) and matching non-lesional (green) fibers are used to calculate the single fiber diffusivity profile (matching points of lesional and non-lesional fibers are connected by white arrows). All points within the lesional core are averaged together and compared to averaged corresponding points of non-lesional fibers. b) Pipeline for calculating single lesion diffusivity profile: panel a:fiber tract intersecting individual lesion is selected, panel b:all individual fibers from selected fiber tract which intersect the lesion and all corresponding non-lesional fibers are selected, panel c:non-lesional fibers are selected for each individual lesional fiber, panel d: diffusivity profiles for each lesional/non-lesional pair are constructed. Right column shows absolute values (blue-diffusivity profile of lesional fibers, red-diffusivity profile of non-lesional fibers. Left column shows diffusivity asymmetry, i.e. difference between lesional and non-lesional fibers, panel e-individual fibers profiles averaged together to calculate single fiber tract diffusivity profile of individual lesion.

lesional fibers were discarded). By using the same perpendicular ROIs to limit non-lesional fibers, the length of lesional/non-lesional fiber pairs included in the analysis was congruent. Similarly, proximity and parallel orientation of fiber pairs was achieved by ensuring that 90% of the voxels of each matched non-lesional fiber was constrained within a 5 mm diameter 'tube' with the lesional fiber at its central longitudinal axis. In order to avoid an effect of Wallerian degeneration, all fibers intersecting with more than 1 lesion were excluded from analysis. In addition, all fibers intersecting with the either the CSF (which was extended by 1 voxel) or grey matter mask were excluded.

Secondly, AD and RD diffusivity profiles were separately calculated for each lesional/non-lesional pair (see examples in Fig 2b, part c). Due to the typically irregular shape of MS lesions, the length of the "intra-lesional" component of lesional streamlines can vary substantially, resulting in length heterogeneity of constructed fiber pairs. Therefore, to standardise measurement of the diffusivity profile, all voxels of individual lesional fibers situated within the lesion core were averaged together to generate a single diffusivity value for the lesion core. Similarly, averaging was performed for corresponding voxels of matching non-lesional fibers to produce a single non-lesional reference diffusivity value. As a result, each lesional/non-lesional pair had length of 13 points (1 point representing lesion core plus 1 and 5 points on each side corresponding to lesion rim and extra-lesional part respectively) (Fig 2a). Examples of AD and RD diffusivity profiles for an individual lesional/non-lesional pair are presented in Fig 2b, part d, left column.

Thirdly, ΔAD and ΔRD profiles for each single fiber were computed by calculating the difference (asymmetry) between diffusivity measures along the selected individual lesional fiber and corresponding points of averaged diffusivity of the matching non-lesional fibres on a voxel-by-voxel basis (Fig 2b, part e, right column).

All single fiber profiles intersecting individual lesions were combined into AD and RD lesional, non-lesional and asymmetry (ΔAD and ΔRD) profiles for each lesion. Patient-wise diffusivity profiles were then computed as a weighted average of the diffusivity profiles of all individual lesions (Fig 3). The weighting (which was performed to adjust for difference in lesion size) was proportionally related to the total number of fibers in all the lesions. Since ΔAD and ΔRD profiles were highly symmetrical with respect to the lesion core, for the purpose of further analysis corresponding points of ΔAD and ΔRD profiles on both sides of the lesion were averaged together, producing a single value of ΔAD and ΔRD for points equally removed from the lesion core (so called Personalized Lesional Diffusivity (PLD) Profile) (Fig 3).

## Control for crossing fibers

MD represents direction-insensitive measure of the total fiber membrane density [21]) and, as a result, remained relatively uniform across the entire white matter in normal brain tissue (see

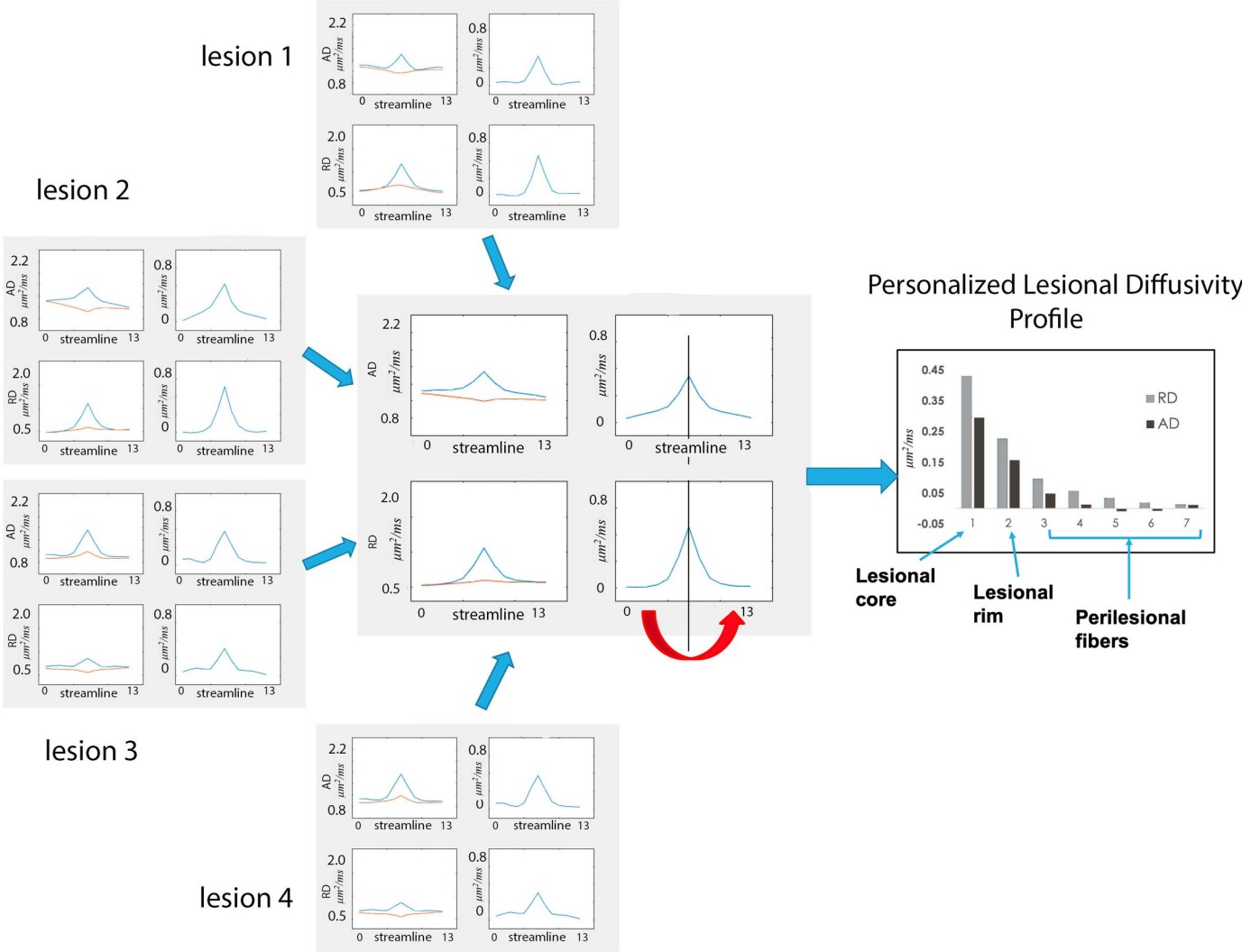

**Fig 3. Computation of personalized lesional diffusivity profile.** Diffusivity profiles of individual lesions are averaged to produce a patient-specific profile. Corresponding points of the left and right parts of the asymmetry profile are averaged together (red arrow), producing a single value of ΔAD and ΔRD for points equally removed from the lesion core (Personalized Lesional Diffusivity (PLD) Profile-see graph on the right). Vertical axis—μm²/ms.

S1a Fig). Conversely, AD and RD, in accordance with their nature, are strongly dependant on fiber coherence. As mentioned above, this represents a serious impediment to the measurement of pathology-related AD and RD changes. However, several features of the proposed technique might minimise the effect of crossing fibers.

Thus, corresponding voxels, which are used to analyse asymmetry of lesional and matching non-lesional fibers of each individual fiber tract have, by definition, at least one common set of fiber bundles, reducing, therefore, potential difference in fiber orientation between the two. In addition, owing to the close proximity of lesional and matching non-lesional fibers the effect of crossing fibers on diffusivity of the corresponding pair of voxels is likely to be similar and, hence, minimized when the difference between the two (i.e. asymmetry) is calculated. Furthermore, adding together large numbers of individual lesional/non-lesional pairs of fibers belonging to different fiber tracts and, therefore, crossing corresponding voxels in different directions is also likely to average out the effect of crossing fibers, at least to some extent.

It is, however, still possible that crossing fibers from other tracts may partially or fully intersect one group of voxels, but not another (for instance, intersect only voxels which belong to lesional fibers, but not voxels associated with non-lesional fibers). This will result in significant change of AD and RD values in voxels containing additional crossing fibers, "contaminating", therefore, the measurement of tissue damage.

The direction of AD and RD change in such a case, however, is expected to be opposite. Thus, presence of additional crossing fibers will result in reduction of the fiber coherence within the voxel, which will cause decrease of diffusivity along the main axis (i.e. reduction of AD) and increase of diffusivity in other directions (i.e. rise of RD) compare to corresponding voxel not affected by the crossing fibers. To validate this assumption, we examined association between AD and RD in individual voxels of normal white matter, which indeed demonstrated strong negative relationship (r = 0.85, p<0.001, see S1a and S1b Fig).

Furthermore, while MD is typically elevated in MS lesions due to increased amount of interstitial fluid and reduction of axonal membranes and myelin caused by axonal loss, comparable level of MD between corresponding voxels which belong to lesional and non-lesional streamlines outside the lesion would indicates similar degree of tissue preservation, suggesting that all relative changes of AD and RD in those voxels are due to variations in fiber coherency (i.e. crossing fibers).

This reasoning was used to detect and minimise the potential effect of crossing fibers on AD and RD in corresponding voxels of lesional and non-lesional streamlines by using custom-designed algorithm implemented in Phyton (see S1 File).

### Statistical analysis

Statistical analysis was performed using SPSS 22.0 (SPSS, Chicago, IL, USA). Pearson correlation coefficient was used to measure statistical dependence between two numerical variables. $P < 0.05$ was considered statistically significant. Comparisons between groups were made using Student $t$-test. Shapiro-Wilk test was used to test for normal distribution.

## Results

### Diffusivity profile of chronic MS lesions

Single fiber-based diffusivity analysis was performed in 30 RRMS patients (age: 43.6+/-9.9 years, EDSS: 1.4+/-1.2, disease duration: 5.3+/-3.5 years, m/f ratio: 11/19). In total 314 lesions (average lesion volume 760 mm$^2$) were analysed.

Examples of patient-based diffusivity profiles and the averaged (across all patients) diffusivity profile are shown in Fig 4. Average number of fibers per lesional voxel was 82+/-71.

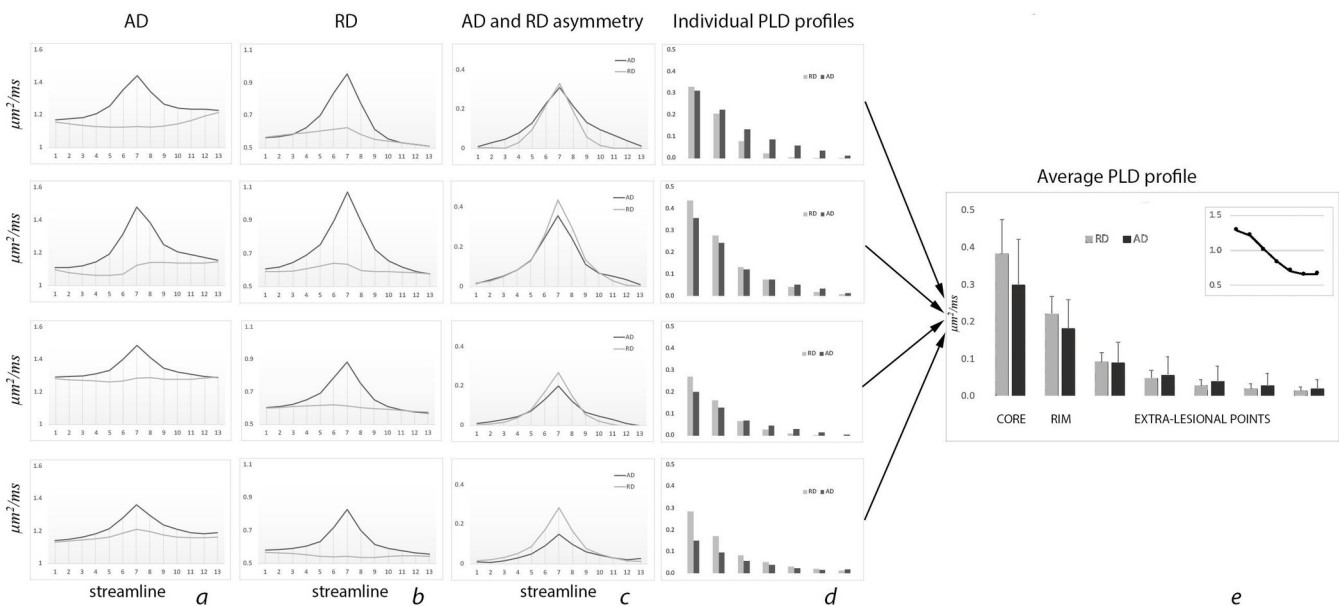

**Fig 4. PLD profile.** a. Individual examples of AD profile (dark line-diffusivity of lesional fibers, grey line-diffusivity of non-lesional fibers). b. Individual examples of RD profile (dark line-diffusivity of lesional fibers, grey line-diffusivity of non-lesional fibers). c. Individual examples of AD and RD asymmetry profile (dark line-AD, grey line-RD). d. Individual examples of PLD profile (dark bars-AD, grey bars-RD). e. Average diffusivity profile. Insert—relative magnitude of $\Delta$AD and $\Delta$RD along the PLD profile. Vertical axis- $\mu m^2/ms$.

Columns *a* and *b* display typical examples of lesional and non-lesional AD (column *a*) and RD (column *b*) profiles, while columns *c* shows AD and RD asymmetry profiles ($\Delta$AD and $\Delta$RD). Individual PLD profiles presented in column *d*.

Patient-based diffusivity profiles of lesional fibers demonstrated a significant increase of both RD and AD in the lesion core compared to non-lesional fibers (average $\Delta$RD and $\Delta$AD 0.38+/-0.09 $\mu m^2/ms$ and 0.30+/-0.12 $\mu m^2/ms$, p<0.0001 for both) that gradually and symmetrically (on both sides) diminished away from the lesion, in some patients almost reaching the level of diffusivity in non-lesional fibers (Table 1). However, both RD and AD of lesional fibers outside the lesion remained significantly higher than diffusivity measures in the corresponding points of non-lesional fibers (paired t-test, p<0.0001 for all points). There was a high correlation (r = 0.67, p<0.001) between relative asymmetry of AD within and outside of lesions (lesion core vs average of 2 most remote extra-lesional points). However, the correlation of lesional $\Delta$RD with RD asymmetry outside of lesion was not significant (p = 0.07).

The relative magnitude of $\Delta$AD and $\Delta$RD varied considerably along the PLD profile. While inside the lesion (both in the core and in the rim) averaged $\Delta$RD was larger than $\Delta$AD, this relationship was reversed in the extra-lesional part of PLD profile, where $\Delta$AD dominated.

**Table 1. Averaged diffusivity values (mean (SD)) in lesion core, rim and extra-lesional part of lesional fibers and in corresponding voxels of matching non-lesional fibers.** (units-$\mu m^2/ms$).

| | Core | Rim | Extra-lesional part (distance from lesion) | | | | |
|---|---|---|---|---|---|---|---|
| | | | 1 mm | 2 mm | 3 mm | 4 mm | 5 mm |
| AD lesion fibers | 1.52 (0.14) | 1.40 (0.09) | 1.3 (0.08) | 1.27 (0.08) | 1.26 (0.08) | 1.25 (0.07) | 1.24 (0.07) |
| AD non-lesion fibers | 1.22 (0.08) | 1.22 (0.07) | 1.21 (0.07) | 1.22 (0.07) | 1.22 (0.06) | 1.22 (0.06) | 1.22 (0.06) |
| RD lesion fibers | 1.0 (0.12) | 0.82 (0.06) | 0.68 (0.05) | 0.63 (0.05) | 0.61 (0.05) | 0.59 (0.05) | 0.59 (0.05) |
| RD non-lesion fibers | 0.61 (0.05) | 0.60 (0.05) | 0.59 (0.05) | 0.59 (0.05) | 0.58 (0.05) | 0.58 (0.05) | 0.57 (0.04) |

This is demonstrated in Fig 4e (see Insert), which shows ΔAD/ΔRD ratio at different points along the PLD profile.

## Relationship between T1 hypointensity and diffusivity profile in the core of chronic MS lesions

Previous studies have demonstrated that the degree of hypointensity on T1-weighted MR images (T1 hypointensity) reflects the severity of axonal loss in MS lesions [22, 23]. Therefore, we used T1 hypointensity to estimate the degree of axonal damage in chronic lesions. Overall, 33% of all lesional voxels contained at least one streamline and were used to analyse T1-hypointensity. Normalized (by NAWM) reduction of intensity of the T1 signal was significantly more prominent (by 26.8%) in voxels used for the analysis compared to the rest of the lesion (reduction of T1 intensity 592 and 751 units respectively, p<0.001, paired t-test), indicating that voxels with better axonal preservation have higher chance of containing tractography streamlines and, therefore, are preferentially selected for single fiber-based analysis. This was also supported by the significant negative correlation between the proportion of analysed voxels in each lesion and lesion T1 hypointensity (r = -0.53, p = 0.002).

Based on assumptions that NAWM represents intact (i.e. no axonal loss) brain tissue (average T1 intensity value of NAWM was 3326 units), while CSF characterises brain tissue totally devoid of axons (average value of minimum T1 intensity of CSF = 863 units) and the intensity of T1 signal is inversely proportional to the degree of axonal loss, we estimated the percentage of patient-wise tissue loss. According to this calculation, the loss of axonal tissue in the lesion core ranged between 22% and 66% (average 36.1+/-11.6%).

In order to examine a link between axonal loss and an increase of parallel and perpendicular diffusivity measures derived using the single-fiber approach, patient-based correlation between degree of axonal loss (as defined above) and ΔAD and ΔRD in the core and the rim of chronic lesions was performed. The analysis of lesion core revealed a high degree of correlation between increase in AD and axonal loss in corresponding voxels (r = 0.82, p<0.0001). While increase in RD in the lesion core was also significantly associated with loss of T1 signal, the correlation was moderate (r = 0.60, p<0.0001). The slope of ΔAD correlation with axonal loss was considerably steeper than the slope of ΔRD (0.0034 vs 0.0021).

Furthermore, the trendline of the ΔAD vs T1-hypointensity derived axonal loss function intersected both axes at zero, indicating close agreement between two measures in assessing the axonal loss (Fig 5a, arrow). Conversely, the trendline of the ΔRD function demonstrated high positive value (0.20 μm$^2$/ms) (Fig 5b, arrow) at zero axonal loss. The presence of a significant residual value of ΔRD suggests that even lesions with preserved axonal content (i.e. T1-isointense lesions) exhibit a significant increase in RD.

In agreement with previous reports, diffusivity was lower in the lesion rim than the lesion core (Table 1). The degree of axonal loss (as determine by T1-hypointensity) in the voxels corresponding to the rim area was also reduced compared with the lesion core (21.2+/-6.0%, range 13–34%). Correlation between T1 hypointensity-derived axonal loss and ΔAD, however, remained high (r = 0.73, p<0.0001). The trendline of the correlation function demonstrated a similar slope to the lesion core (0.0035) and intersected with the zero value of the horizontal (T1-hypointensity derived axonal loss) axis very close to zero value of ΔAD (-0.004 μm$^2$/ms) (Fig 5c), again indicating close agreement between T1 signal and ΔAD as a measure of axonal damage. Conversely, ΔRD was only weakly correlated with T1 signal (r = 0.39, p = 0.026) showing flatter slope of the correlation curve compare to the correlation in the lesion core (0.0013). However, intersection of the ΔRD trendline with zero value of the axonal loss axis still demonstrated significant residual value of ΔRD (0.15 μm$^2$/ms) (Fig 5d).

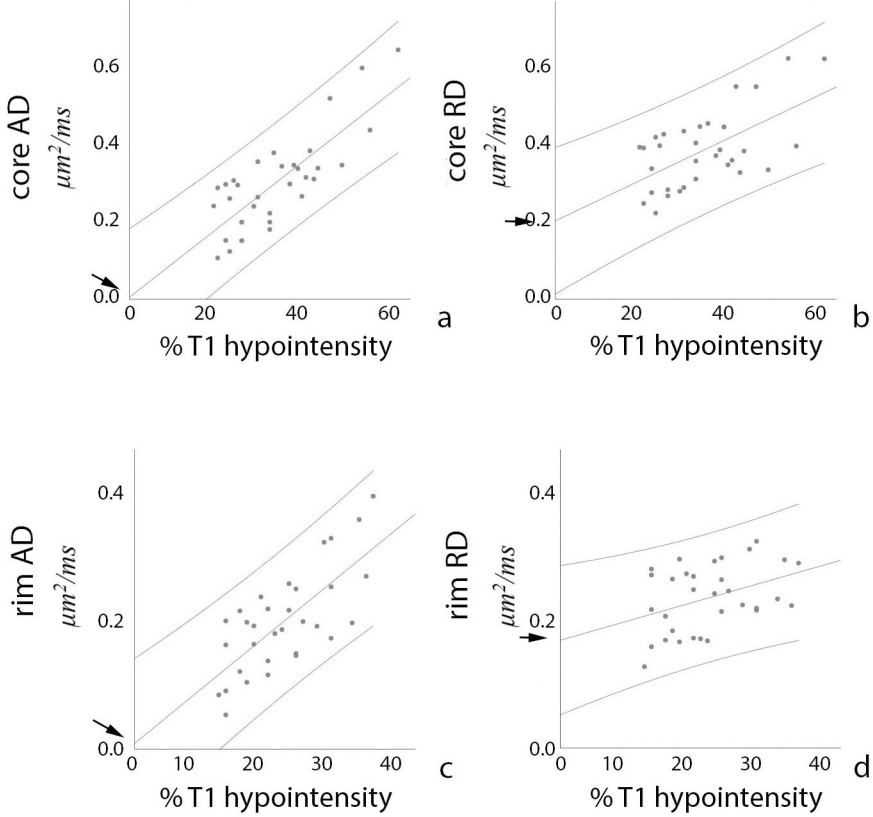

**Fig 5. Correlations between diffusivity increase (vertical axis) and T1 hypointensity-derived axonal loss (horizontal axis).** Arrows point at intersection between diffusivity increase and "zero" axonal loss (p values are noted a text). Vertical axis—μm²/ms, Horizontal axis—% of T1 hypointensity-derived axonal loss.

By further examining changes of AD and RD in the lesional core of individual patients, we observed considerable variation in both the magnitude of diffusivity increase (reflected in a relatively large standard deviation, Fig 4d), and the relationship between ΔAD and ΔRD. Thus, in some patients ΔRD in the lesional core was significantly larger than ΔAD (see, for example Fig 4, bottom row), while in other patients a similar increase of parallel and perpendicular diffusivities was noted (Fig 4, top row). Therefore, we examined how the difference between the change of radial and axial diffusivities (ΔRD-ΔAD) is related to the degree of axonal loss. The result, shown in Fig 6, demonstrates significant negative correlation between the level of preferential RD increase and the degree of (T1 hypointensity-derived) axonal destruction (r = -0.62, p<0.001), indicating that ΔRD dominates in cases with milder axonal loss (where majority of axons still survive, but remain demyelinated), but ΔAD and ΔRD approach parity as axonal damage advances.

## Modelling effects of axonal loss and demyelination on diffusivity in the lesion core

Our previous work suggests that, in highly coherent fiber tracts, ΔAD within chronic MS lesions reflects axonal loss, while ΔRD is related to both axonal loss and demyelination in survived axons [24]. Since the current approach substantially minimizes the effect of crossing fibers, we hypothesized that diffusion modelling within coherent fiber tracts can be extended

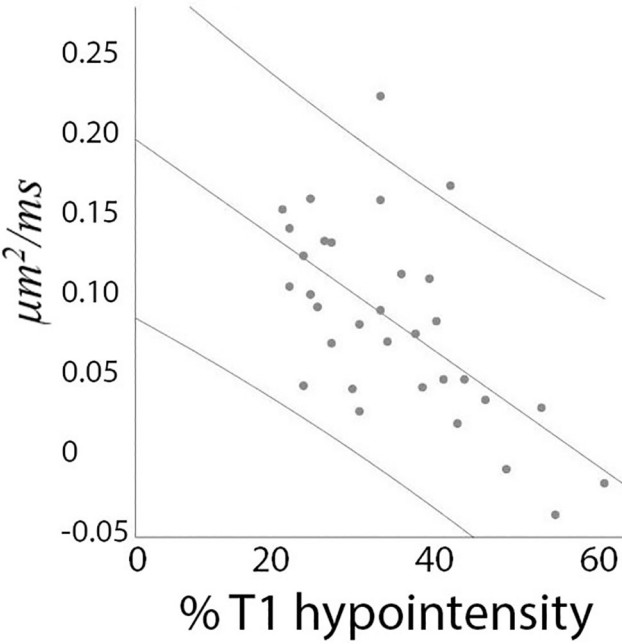

**Fig 6. Correlation between ΔRD-ΔAD (vertical axis) and T1 hypointensity-derived axonal loss (horizontal axis).** Vertical axis—μm²/ms, Horizontal axis—% of T1 hypointensity-derived axonal loss.

to lesions involving any region of the brain. In the current study, we found a high degree of correlation between T1 hypointensity-defined axonal loss and ΔAD and congruence of the "zero" value of both measures (based on trendline projection), supporting the close relationship between increase in AD and the degree of axonal destruction. Conversely, we found only a moderate correlation of T1 hypointensity with ΔRD and a lesser slope of the correlation curve, indicating that factors not directly associated with axonal degeneration in part drive the increase in perpendicular diffusivity observed in chronic MS lesions (see comparison of ΔAD and ΔRD trendlines in Fig 7a). Furthermore, the high residual value of ΔRD at "zero" axonal loss level and a preferential increase of RD in patients with mild axonal loss (which gradually diminishes as axonal loss advances), suggest that neurodegeneration and non-axonal loss related factors have opposing effects on ΔRD. A declining impact of non-axonal loss related

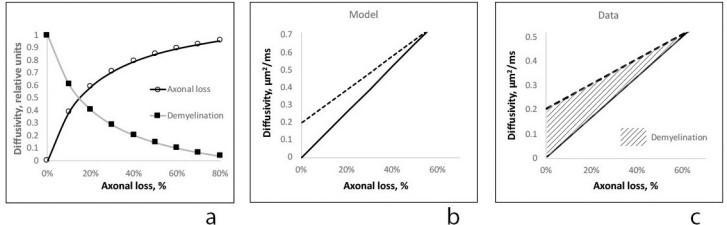

**Fig 7. Results of diffusivity modelling.** a. Relative contribution of demyelination-related ΔRD$^{dem}$ (black squares) and axonal loss-related ΔRD$^{ax}$ (circles) to radial diffusivity increase in core of MS lesions. Vertical axis -relative units, Horizontal axis—T1 hypointensity-derived axonal loss. b. Modeling of ΔRD (dash line) and ΔAD (solid line) vs axonal loss. Vertical axis—μm²/ms, Horizontal axis—% of T1 hypointensity-derived axonal loss. c. Trendlines of ΔRD (dash line) and ΔAD (solid line) vs axonal loss (lesion core). Stripe area represents potential contribution of demyelination to ΔRD. Vertical axis—μm²/ms, Horizontal axis—% of T1 hypointensity-derived axonal loss.

factors on ΔRD as axonal damage become more prominent implicates loss of the myelin sheath in survived (i.e. existing) axons, since the impact of demyelination (which is known to increase RD) is expected to be largest in lesions with relatively well-preserved axons. However, since the number of (demyelinated) axons decline as axonal loss progresses the contribution of demyelination to ΔRD is also expected to decrease proportionally with attrition of demyelinated axons. It is therefore plausible to assume that the increase of RD in chronic MS lesions is driven by the combination of two independent factors: loss of axons and loss of myelin sheath in survived axons. Following this logic, we modelled the relationship between axonal loss, demyelination and changes in AD and RD to evaluate how well our assumption explains the real data.

Based on replacement of lost axonal tissue by extra-cellular water, we previously estimated the increase in axial diffusivity (ΔAD) in chronic MS lesions [24]. Namely,

$$\Delta AD = \{(1 - f) \text{ x } AD^{normal\ tissue}) + (f \text{ x } AD^{ax\ loss})\} - AD^{normal\ tissue}$$

*Where*:

"*f*" *volume fraction (%) of axonal loss*
"*1 –f*" *volume fraction of normal tissue*
$AD^{normal\ tissue}$ *average AD in non-lesional fibers (1.22 μm²/ms)*
$AD^{ax\ loss}$ *AD in lesion with complete axonal loss (2.5 μm²/ms)*

Increase in radial diffusivity was modelled as a sum of demyelination-related $\Delta RD^{dem}$ and axonal loss-related $\Delta RD^{ax}$ changes:

$$\Delta RD = \Delta RD^{dem} + \Delta RD^{ax}$$

Assuming that in hypothetical lesions with fully preserved axons, the increase of RD is entirely driven by demyelination, the intersection of the ΔRD trendline with the zero value of the (horizontal) axonal loss axis (Fig 5b) was used to identify the value of demyelination-related increase of RD ($\Delta RD^{dem}$), which was equal to 0.2 μm²/ms. (This was similar to previously reported demyelination-related increase of RD in highly coherent tracts, such as optic radiation (0.23 μm²/ms)) [24]. Therefore, demyelination-related increase of RD was calculated as follows:

$$\Delta RD^{dem} = (0.20 \text{ x } (``1 - f\,"))$$

Axonal loss-related $\Delta RD^{ax}$ was calculated as a replacement of lost axonal tissue by extra-cellular water:

$$\Delta RD^{ax} = \{(1 - f) \text{ x } RD^{normal\ tissue}) + (f \text{ x } RD^{ax\ loss})\} - RD^{normal\ tissue}$$

*Where*:

$RD^{normal\ tissue}$ *average AD in non-lesional fibers (0.59 μm²/ms)*
$RD^{ax\ loss}$ *RD in lesion with complete axonal loss (1.7 μm²/ms)*

We previously reported that in chronic MS lesions situated within highly coherent fiber tracts the maximum AD and RD values do not exceed 2.5 μm²/ms and 1.7 μm²/ms respectively [24], indicating some degree of diffusion restriction and residual anisotropy even in severely damaged white matter, which was attributed to lesional gliosis and axonal tortuosity [8–10, 25]. Therefore, those values were used in the current modelling, presented in Fig 7.

The relative contribution of two opposing processes, increasing $\Delta RD^{ax}$ and decreasing $\Delta RD^{dem}$, to ΔRD is plotted in Fig 7a, highlighting the leading contribution of demyelination to ΔRD in lesions with minimal axonal loss, which rapidly diminishes as axonal loss progresses. Fig 7b demonstrates the predicted slopes of ΔRD and ΔAD vs axonal loss. While the increase

in RD is larger at low levels of axonal loss, ΔRD and ΔAD become equal as axonal degeneration advances. Note that the similarity of the relationship of ΔRD and ΔAD slopes between the model and experimental data (ΔRD and ΔAD trendlines of lesion core data are presented in Fig 7c).

## Discussion

While DTI was suggested as a potential *in-vivo* biomarker of axonal loss and demyelination in MS more than a decade ago, its practical application, including use in clinical trials, is limited by inherent dependency on fiber coherency and complexity of the underlying brain structure.

In the current work, we describe a novel approach to analyse diffusivity in chronic MS lesions, based on a computation of a difference (asymmetry) between diffusion properties of a single lesional streamline and corresponding points in neighbouring non-lesional streamlines of similar length and orientation selected from the same fiber tract.

The proposed approach was designed to minimise the impact of crossing fibers on the inter-voxel variability of orientation-selective diffusivity measures and to eliminate the confounding effect of irregular MS lesion morphology, thereby improving the detection of pathology-specific DTI changes in MS lesions.

We observed a parallel change in lesional and matching non-lesional AD and RD profiles external to the lesion, as seen in individual examples (Fig 4); and symmetrical (on either side of the core) alteration in RD and AD (Fig 4, column c), supporting our proposition that single fiber-based technique can potentially extend the application of diffusivity profile analysis beyond coherent fiber tracts, which, we believe, is facilitated by the close proximity of corresponding lesional and non-lesional voxels, contribution of parallel fibers from at least one common fiber tract, averaging of a large number of differently oriented pairs of lesional/non-lesional fibers for each set of corresponding voxels and normalisation algorithm.

While the analysis of lesional vs non-lesional fibers is performed on a voxel-based basis and, therefore includes contribution from all fibers crossing the voxel in different directions (i.e. diffusivity of the entire voxel which belongs to the particular point of the lesional fiber is compared to the diffusivity of the entire voxel which belongs to the corresponding point of the non-lesional fiber), the advantage of the proposed technique is that it provides a true "local" reference for the measurement of pathology-related diffusivity changes inside MS lesions and in its immediate surrounding.

The single fiber-based approach also obviates the confounding effect of irregular lesion morphology on traditional tract-based analyses. Due to the typically irregular shape of MS lesions, the length of the "intra-lesional" component of individual streamlines can vary substantially. Therefore, averaging of all lesional fibers (even within single fiber tract) and comparison with averaged non-lesional fibers, as is typically implemented in tract-based analysis, will inevitably result in significant distortion of the lesional diffusivity profile. Since each lesional streamline is individually matched to non-lesional fibers of similar length in the approach proposed here, the "intra-lesional" part each lesional fiber is identified independently and asymmetry analysis of each individual lesional/non-lesional pair performed prior to averaging. As a result, only corresponding parts of the asymmetry profile are averaged, eliminating the effect of irregular lesion shape on calculation of the diffusivity profile.

We used the single fiber-based approach to examine orientation-sensitive diffusivity measures (AD and RD) in chronic white matter lesions of MS patients. Our analysis revealed a substantial increase of both AD and RD in the lesion core, which diminished towards the lesion rim. While small, altered diffusivity remained significant even in the extra-lesional component of lesional fibers. Considerable variability of the increase of lesional diffusivity was observed

between patients, although the magnitude of ΔRD was typically larger. Parallel and perpendicular diffusivities also demonstrated different behaviour in relation to axonal loss. In particular, there was high degree of correlation between increased axial diffusivity and T1 hypointensity-based measures of tissue destruction in the lesion core, suggesting a strong association between ΔAD and axonal loss. Furthermore, the trendline of the correlation function intersected both axes at zero, reinforcing the notion that both measures reflect similar underlying patho-mechanisms. Since T1-hypointensity is linked to the degree of axonal destruction, it is reasonable to assume that ΔAD derived using the single fiber-based approach is also strongly associated with axonal loss in MS lesions.

The correlation of ΔRD with T1-hypointensity in the lesion core, however, was only moderate. Moreover, contrary to AD, the trendline of the ΔRD correlation function had a flatter slope and intersected zero on the horizontal (T1-hypointensity) axis at a relatively high positive value of ΔRD. In addition, a relatively larger increase of RD (ΔRD-ΔAD) in the lesional core was found to be inversely associated with the degree of T1 hypointensity, indicating that magnitude of ΔRD is higher in cases with milder axonal loss, but ΔAD and ΔRD approach parity as axonal damage advances.

Taken together, our results strongly support a direct link between the increase of axial diffusivity derived using the single fiber-based technique and the degree of axonal destruction. More importantly, the findings support the notion that the increase of RD in chronic MS lesions is driven by the combination of two factors: axonal destruction and loss of myelin sheath in survived axons. While the effect of demyelination dominates in cases of mild axonal destruction (where majority of axons still survive, but remain demyelinated), neurodegeneration is a primary cause of RD increase in more destructive lesions.

Modelling diffusivity changes in the core of chronic MS lesions based on the direct proportionality of ΔAD with axonal loss and the proposed dual nature of ΔRD yielded results that were strikingly similar to the experimental data. The difference in magnitude of the diffusivity increase between the model and the experimental data, noticeable at the high end of the axonal loss scale, is likely to be related to the restricting effect of glial fibers on the diffusion of water molecules [8, 25], which are particularly dense in severely damaged tissue.

The proposed single fiber-based technique preserves the advantage of the fiber-based approach by estimating the diffusivity only in parts of the lesion characterised by relative sparing of axons. Contrary to ROI-based analysis (where all lesional voxels are combined together), single-fiber analysis is "biased" towards the voxels with better axonal preservation since they have relatively high anisotropy and, therefore, a greater chance of containing tractography streamlines. Conversely, voxels with a high degree of axonal loss, which typically exhibit more isotropic diffusivity, are less likely to be included in the current analysis. This is supported by the significantly greater T1 signal in voxels selected for single fiber-based analysis relative to the remaining lesional voxels and a significant negative correlation between the proportion of analysed voxels and whole lesion T1 hypointensity. As a result, the single fiber-based method minimizes the "diluting" effect of existing tissue loss on longitudinal monitoring of orientation-sensitive diffusivity measures such as RD. This is particularly applicable to measurement of remyelination in MS lesions, where remyelination-induced change of diffusivity can only occur in survived (existing) axons.

We recently reported an elevation of AD in the distal part of the lesional fibers in highly coherent fiber tracts (optic radiation) [17]. The distribution of the observed increase in AD suggested that it may be related to the extent of the axonal transection within the lesion and the subsequent loss of a distal part of connected axons caused by Wallerian Degeneration (WD). In concordance with these findings, we observed a highly significant correlation of ΔAD (but not ΔRD) between the lesion core and the extra-lesional part of the lesional fibers

and a preferential increase of AD in lesional fibers outside the lesion in the current study, supporting ΔAD as a potential biomarker of WD, even outside of highly coherent fiber tracts.

Finally, while changes of diffusivity in the lesion rim demonstrated similar trends, the relative increase of both AD and RD was smaller compared to the lesion core. We also observed a higher degree of variability of ΔRD unrelated to axonal loss (as indicated by a weaker correlation with T1 signal) and lower value of ΔRD at the intersection of the trendline with "zero" on the T1 hypointensity axis. Differences from the lesion core are likely be related to less severe axonal loss and a variable degree of remyelination between patients, both characteristic features of the lesion edge. This may render the lesion rim a better target for monitoring subtle changes of treatment-induced myelination in clinical trials [26].

The single fiber-based method has some limitations. While the proposed approach is beneficial for longitudinal analysis of changes affecting survived axons (such as remyelination), voxel-based analysis may be more suited for measurement of tissue loss in cross-sectional comparison studies, especially when axonal loss is severe. Another limitation is related to potential partial voluming effect caused by resampling of DTI images.

Monitoring remyelination using this technique is also limited to lesions with relatively well-preserved axons, since in cases where neurodegeneration exceeds 30–40% ΔRD is mainly driven by axonal loss and the contribution of demyelination becomes negligible. This limitation is probably applicable to all diffusion-based studies of remyelination.

In addition, due to strict constraints imposed by the technique on the location of non-lesional fibers (such as specific distance from the analysed single streamline), more centrally located voxels in large lesions are less likely to be included in analysis.

## Supporting information

**S1 Fig. MD, AD and RD values in normal white matter.** a. Diffusivity values in 200 individual voxels randomly selected in brain's white matter of normal subject. While AR and RD vary significantly (Coefficient of variability: AD-21%, RD-26%) and in opposite directions, MD remains relatively constant (Coefficient of variability: MD-7%). RD values multiplied by 2. b. Correlation between AD and RD in individual voxels randomly selected in brain white matter of normal subject (r = 0.85, p<0.001).
(DOCX)

**S2 Fig. MD-based normalisation.** a. Left column demonstrates original AD and RD profiles of lesional (blue) and non-lesional (red) fibers. Blue arrow indicates linear fitting of lesional fiber. Red arrow indicates linear fitting of non-lesional fibers. Horizontal axis: points along the fibers. Point 7 indicates lesion. Vertical axis: $\mu m^2$/ms. b. Slopes of ΔAD, ΔRD and ΔMD in individual lesional/non-lesional pair.
(DOCX)

**S1 File.**
(DOCX)

## Author Contributions

**Conceptualization:** Samuel Klistorner, Michael H. Barnett, Con Yiannikas, Joshua Barton, John Parratt, Yuyi You, Stuart L. Graham, Alexander Klistorner.

**Data curation:** Samuel Klistorner, Alexander Klistorner.

**Formal analysis:** Samuel Klistorner, Jakob Wasserthal, Alexander Klistorner.

**Funding acquisition:** Alexander Klistorner.

**Investigation:** Samuel Klistorner, Joshua Barton, Alexander Klistorner.

**Methodology:** Samuel Klistorner, Jakob Wasserthal, Alexander Klistorner.

**Project administration:** Alexander Klistorner.

**Resources:** Michael H. Barnett, Alexander Klistorner.

**Software:** Samuel Klistorner, Alexander Klistorner.

**Supervision:** Alexander Klistorner.

**Validation:** Samuel Klistorner, Alexander Klistorner.

**Visualization:** Samuel Klistorner, Alexander Klistorner.

**Writing – original draft:** Alexander Klistorner.

**Writing – review & editing:** Michael H. Barnett, Yuyi You, Stuart L. Graham, Alexander Klistorner.

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
