## [Decision Letter · Decision Letter 0]

5 Nov 2020

PONE-D-20-27494

Differentiating axonal loss and demyelination in chronic MS lesions: a novel approach using single streamline diffusivity analysis.

PLOS ONE

Dear Dr. Klistorner,

Thank you for submitting your manuscript to PLOS ONE. After careful consideration, we feel that it has merit but does not fully meet PLOS ONE’s publication criteria as it currently stands. Therefore, we invite you to submit a revised version of the manuscript that addresses the points raised during the review process. Both expert reviewers found important concerns to be addressed, mainly regarding description of Methods and protocols, as well as in the quality and organization of figures.

We look forward to receiving your revised manuscript.

Kind regards,

Fernando de Castro

Academic Editor

PLOS ONE

Journal Requirements:

3.  Thank you for stating the following in the Financial Disclosure : 

'AK-National Multiple Sclerosis Society (NMSS), Novartis Save Neuron Grant, Sydney Eye Hospital foundation grant and Sydney University Medical foundation.

We note that one or more of the authors have an affiliation to the commercial funders of this research study : 'Novartis'.

Reviewers' comments:

Reviewer's Responses to Questions

**Comments to the Author**

1. Is the manuscript technically sound, and do the data support the conclusions?

Reviewer #1: Yes

Reviewer #2: Partly

2. Has the statistical analysis been performed appropriately and rigorously? 

Reviewer #1: Yes

Reviewer #2: I Don't Know

3. Have the authors made all data underlying the findings in their manuscript fully available?

Reviewer #1: Yes

Reviewer #2: Yes

4. Is the manuscript presented in an intelligible fashion and written in standard English?

Reviewer #1: Yes

Reviewer #2: Yes

5. Review Comments to the Author

Reviewer #1: In this study authors developed a new method for evaluating axonal damage inside MS lesion on brain MRI in 30 MS cases by using a single streamline diffusivity analysis. They found a significant increase of both axial and radial diffusivity and AD showed a very high correlation with T1 hypointensity. This article is a new addition to the contribution of the team to the application of DWI to the understanding on MS lesions and quantification of axonal loss. The team is very well experienced and recognized and this new method is highly valuable for improving our understanding about how MS damage the brain and for testing new neuroprotective and remyelinating therapies.

Comments

1. Methods: indicates the number of voxels that 100 mm3 represent in this specific scanner. Explain how TractSeg algorithm solved the interruption of the tract within the lesions to provide the reconstruction of the tract. Authors explained that they use reference fibbers, but the algorithm should make use of a statistical or probability analysis to approach the fibbers within the lesion. Or it was always required a 100% match with the reference fibber? Indicate how many fibbers on average were present in the DWI voxels.

2. Can you provide the DTI protocol? I was unable to find it in the supplementary material

3. Figures are of low resolution (specially the insets with the graphs). Please, provide high quality figures

4. Figure 5 and 6: provide legends to the X and Y axis as indicated in the figure legend

Reviewer #2: The authors attempt to develop a single-streamlined approach to analyze diffusivity within chronic MS lesions. While the method seems potentially promising there are too many issues with the presentation and interpretation of the data to know if this method is sound.

Major issues:

• In the second last paragraph of the introduction, the authors state, “based on an assumption of uniform mean diffusivity of the brain white matter,” Why is that assumption valid?

• The authors state in the methods section about the MR parameters that “Specific parameters are presented in Supplementary material.” I do not see these parameters in the supplemental material. The administration of Gd can change metrics so I also expected to see the order in which the images were collected and that is not there either.

• Axis labels on the graphs in Figure 2 are necessary. What is the x-axis? What do the blue and the orange lines mean? The titles of the graph should be the y-axis labels. Is difference AD the delta-AD written in the manuscript? The authors should use consistent notation. In Figure 4 they seem to call this AD and RD asymmetry. Consistency is necessary. Labelling also applies to Figure 3-7.

• The bar graph in Figure 3 is also very confusing. Do the bars not represent Delta-RD and Delta-AD? If they do, then the legend needs to be changed. If they do not, a much better explanation of what is in the graph needs to be made. Using blue and orange is also VERY confusing. In the other graphs in Figure 3 and in the graphs in Figure 2, blue is meant to represent the lesion while orange is meant to represent the non-lesion tracks. Yet in the bar graph in Figure 3, blue and orange mean something else. The colors need to be changed for consistency. I am also unclear if I understand things correctly. Do only 2 voxels worth of data go into the lines in 2-7? Are data from a whole lesion core (line 1) really comparable with data from a single voxel ring around the lesion? It seems like the statistics would be poor for the analysis outside the lesion core. The lesion rim likely suffers inhomogeneously from partial voluming effects. Is that considered in the analysis?

• The author’s explanation of their new change in the way data are analyzed normalizing the differences as done in Figure 2 of the supplement is interesting but not convincing. What other factors can result in a change in RD and AD but not MD and have the authors checked to see if those effects are seen in their work and how they would affect their work. For example, some work has been done indicating axon diameter changes in different area of the brain. I would think that could result in a consistent MD but a change in AD and RD. From what the authors have written, they appear to have assumed that the changes they see are from crossing fibers and not from some other means. I need to be convinced they are not from other means or that these other means are irrelevant.

• Figure 4 switches the color scheme for Figure e. a-d had AD dark and RD light and Figure e has RD dark and AD light. Consistency is needed.

• Table 1: what are the uncertainties in the numbers in the table?

Minor issues:

• Very minor English mistakes can be found throughout the manuscript (for example “this represent”. A quick read though to fix these mistake should be done.

• Page 9 of the document, paragraph 4, the authors refer to “normal” cohorts. I believe the disease societies are asking scientists to stop using these terms and rather use terms such as “non-MS” cohorts.

6. PLOS authors have the option to publish the peer review history of their article (what does this mean?). If published, this will include your full peer review and any attached files.

Reviewer #1: **Yes: **Pablo Villoslada

Reviewer #2: No

---

## [Author Response · Author response to Decision Letter 0]

23 Nov 2020

Please find response to reviewers’ comments below. 

RReviewer #1: In this study authors developed a new method for evaluating axonal damage inside MS lesion on brain MRI in 30 MS cases by using a single streamline diffusivity analysis. They found a significant increase of both axial and radial diffusivity and AD showed a very high correlation with T1 hypointensity. This article is a new addition to the contribution of the team to the application of DWI to the understanding on MS lesions and quantification of axonal loss. The team is very well experienced and recognized and this new method is highly valuable for improving our understanding about how MS damage the brain and for testing new neuroprotective and remyelinating therapies.

Comments

Q 1. Methods: indicates the number of voxels that 100 mm3 represent in this specific scanner. Explain how TractSeg algorithm solved the interruption of the tract within the lesions to provide the reconstruction of the tract. Authors explained that they use reference fibbers, but the algorithm should make use of a statistical or probability analysis to approach the fibbers within the lesion. Or it was always required a 100% match with the reference fibber? Indicate how many fibbers on average were present in the DWI voxels.

A. 100 mm3 represent 100 voxels on T2 FLAIR image. Now added to Method section.

TractSeg only generates binary tract masks, which, however, makes it easy to generate streamlines since seeding for probabilistic tractography is produces within the mask and certain number (2000) of streamlines are required for each tract. The TractSeg methodology through MS lesion is described in 1. This is now added in Method section

Reference fiber was always required. Lesional fibers which did not have matching reference “non-lesional” fibers were not included in analisys.

Average number of fibers per lesional voxel: 82+/-71 (added to Result section.)

Q. 2. Can you provide the DTI protocol? I was unable to find it in the supplementary material

A. Sorry MRI protocol was not included. Added now in Suppl material. 

Q. 3. Figures are of low resolution (specially the insets with the graphs). Please, provide high quality figures

A.All figures are now 300p resolution. 

Figures 2b, 3, 5 are improved

Q 4. Figure 5 and 6: provide legends to the X and Y axis as indicated in the figure legend

A.Legend for fig 5 and 6 are now provided

Reviewer #2: The authors attempt to develop a single-streamlined approach to analyze diffusivity within chronic MS lesions. While the method seems potentially promising there are too many issues with the presentation and interpretation of the data to know if this method is sound.

Major issues:

Q • In the second last paragraph of the introduction, the authors state, “based on an assumption of uniform mean diffusivity of the brain white matter,” Why is that assumption valid?

A.This assumption is based on our extensive experience of observing Mean Diffusivity images, which generally look homogeneous (contrary to AD or RD maps). This is further quantified in Supplementary material, (see Suppl Fig 1 which demonstrated very low variability of MD values across white matter-coef of variability 7%). However, we agree with reviewer that presenting this as an assumption is not justified. Therefore, sentence is modified as follow ”…based on an observation that distribution of mean diffusivity in the brain white matter is relatively uniform (see Suppl material, Fig.1)”

Q • The authors state in the methods section about the MR parameters that “Specific parameters are presented in Supplementary material.” I do not see these parameters in the supplemental material. The administration of Gd can change metrics so I also expected to see the order in which the images were collected and that is not there either.

A.Sorry, MRI protocol was not included. Added now in Suppl material. Order of image acquisition is also added. 

Q • Axis labels on the graphs in Figure 2 are necessary. What is the x-axis? What do the blue and the orange lines mean? The titles of the graph should be the y-axis labels. Is difference AD the delta-AD written in the manuscript? The authors should use consistent notation. In Figure 4 they seem to call this AD and RD asymmetry. Consistency is necessary. Labelling also applies to Figure 3-7.

A.We thank reviewer for the suggestion. Labels are added to Fig.2 In addition, following description is added to figure legend “Right column shows absolute values (blue-diffusivity profile of lesional fibers, red- diffusivity profile of non-lesional fibers. Left column shows asymmetry between lesional and non-lesional fibers”, i.e. delta AD or RD”.

As suggested, “difference” changed to “asymmetry”, for consistency.

Fig. 3-7 modified as suggested.

Q • The bar graph in Figure 3 is also very confusing. Do the bars not represent Delta-RD and Delta-AD? If they do, then the legend needs to be changed. If they do not, a much better explanation of what is in the graph needs to be made. Using blue and orange is also VERY confusing. In the other graphs in 

A.Reviewer is correct, bars in Fig. 3 represent Delta-RD and Delta-AD. This is explained in Method section: “�AD and �RD for points equally removed from the lesion core called Personalized Lesional Diffusivity”.

Q Figure 3 and in the graphs in Figure 2, blue is meant to represent the lesion while orange is meant to represent the non-lesion tracks. Yet in the bar graph in Figure 3, blue and orange mean something else. The colors need to be changed for consistency. 

A.We thank reviewer for the suggestion. Colours corrected.

Q I am also unclear if I understand things correctly. Do only 2 voxels worth of data go into the lines in 2-7? Are data from a whole lesion core (line 1) really comparable with data from a single voxel ring around the lesion? It seems like the statistics would be poor for the analysis outside the lesion core. The lesion rim likely suffers inhomogeneously from partial voluming effects. Is that considered in the analysis?

A.Reviewer is correct in assuming that only 2 voxels are considered for each point between 2 and 7. However, those 2 points applied to SINGLE lesional fiber profile and it is compared to core voxels of SINGLE fiber (which can be anything from 1 to many, depending on size of the lesion). To calculate Personalised Lesional Profile presented in Fig 3But many single fiber profiles (often hundreds) are used. Therefore, statistics for the comparison between core and other layer is sound. Partial volume effect was not considered in the analysis, which is now added to study limitations. 

Q • The author’s explanation of their new change in the way data are analyzed normalizing the differences as done in Figure 2 of the supplement is interesting but not convincing. What other factors can result in a change in RD and AD but not MD and have the authors checked to see if those effects are seen in their work and how they would affect their work. For example, some work has been done indicating axon diameter changes in different area of the brain. I would think that could result in a consistent MD but a change in AD and RD. From what the authors have written, they appear to have assumed that the changes they see are from crossing fibers and not from some other means. I need to be convinced they are not from other means or that these other means are irrelevant.

A.We thank reviewer for the comment. There are indeed some publications suggesting variation of axonal diameter in different parts of the brain However, since our analysis is based on the difference between lesional and neighbouring non-lesional fibers, we believe that this factor is not likely to play significant role in opposite change of AD and RD, used for normalisation. Apart from variability of axonal diameter, mentioned by the reviewer, we cannot think of any other reason for observed phenomenon of stable MD and opposite change of RD and AD except for crossing fibers. 

Q • Figure 4 switches the color scheme for Figure e. a-d had AD dark and RD light and Figure e has RD dark and AD light. Consistency is needed.

A.We thank reviewer for the suggestion. Colours corrected.

Changed as suggested.

Q • Table 1: what are the uncertainties in the numbers in the table?

A.Standard Deviation in presented in brackets

Minor issues:

Q • Very minor English mistakes can be found throughout the manuscript (for example “this represent”. A quick read though to fix these mistake should be done.

A.We thank reviewer for the suggestion. Corrected.

Q • Page 9 of the document, paragraph 4, the authors refer to “normal” cohorts. I believe the disease societies are asking scientists to stop using these terms and rather use terms such as “non-MS” cohorts.

A.We thank reviewer for the suggestion. Corrected as suggested.

---

## [Decision Letter · Decision Letter 1]

8 Dec 2020

PONE-D-20-27494R1

Differentiating axonal loss and demyelination in chronic MS lesions: a novel approach using single streamline diffusivity analysis.

PLOS ONE

Dear Dr. Klistorner,

Thank you for submitting your manuscript to PLOS ONE. After careful consideration, we feel that it has merit but does not fully meet PLOS ONE’s publication criteria as it currently stands. Therefore, we invite you to submit a revised version of the manuscript that addresses the points raised during the review process. The very minor changes suggested by expert reviewer #2 are related to some overstatements in the text, as well as apparent confessions, between the title of figures and the axes in plots. Please, revise them and send them back to us for their final approval.

We look forward to receiving your revised manuscript.

Kind regards,

Fernando de Castro

Academic Editor

PLOS ONE

Reviewers' comments:

Reviewer's Responses to Questions

**Comments to the Author**

1. If the authors have adequately addressed your comments raised in a previous round of review and you feel that this manuscript is now acceptable for publication, you may indicate that here to bypass the “Comments to the Author” section, enter your conflict of interest statement in the “Confidential to Editor” section, and submit your "Accept" recommendation.

Reviewer #1: All comments have been addressed

Reviewer #2: (No Response)

2. Is the manuscript technically sound, and do the data support the conclusions?

Reviewer #1: Yes

Reviewer #2: Yes

3. Has the statistical analysis been performed appropriately and rigorously? 

Reviewer #1: Yes

Reviewer #2: I Don't Know

4. Have the authors made all data underlying the findings in their manuscript fully available?

Reviewer #1: Yes

Reviewer #2: Yes

5. Is the manuscript presented in an intelligible fashion and written in standard English?

Reviewer #1: Yes

Reviewer #2: Yes

6. Review Comments to the Author

Reviewer #1: Authors have addressed properly all my comments and doubts. The figures are now of good quality and the labels have been added to the axis

Reviewer #2: The authors have made substantive changes to the manuscript so that their work has become much clearer. I have two issues, which I believe can easily be corrected, with the manuscript as it stands.

1. The authors have a whole bunch of metrics which have been shown in their studies, or in others, to correlate with various pathologies. Yet, the authors do not do the correlations themselves with the people in this study. The manuscript itself does an excellent job not overstating what was done. The abstract, on the other hand, does overstate what was done which can be dangerous as the abstract is often the only part of a manuscript that is read by researchers who are looking for some trick to describe their own data. Thus, I suggest changing the overreaching statement “Our result demonstrates close association between an increase of AD and the degree of axonal loss and support the notion that the increase of RD in chronic MS lesions is driven by the combination of two factors: axonal destruction and loss of myelin sheaths in survived axons. While the effect of demyelination dominates in cases of mild axonal loss, neurodegeneration is a primary cause of increased RD in more destructive lesions. These finding highlight the importance of selecting appropriate patient cohorts for clinical trials of pro-remyelinating and neuroprotective therapeutics.” to something more accurate. The summary the authors wrote at the end of the manuscript has a nice statement that could be used instead, “Evaluation of lesions in a sizable cohort of MS patients using the proposed method supports the use of �AD as a marker of axonal loss; and the notion that demyelination and axonal loss independently contribute to the increase of RD in chronic MS lesions.”

2. Once again the plots, while much more clear, are not in the standard format for plots. The authors seem to be confusing the title of the plot with what should be on the y-axis. This is true for almost every plot in the figures presented. To make things clear, I will explain what I mean using just the top graph of Figure 2b. There should be no title. The y-axis should be labelled AD (µm2/ms). The x-axis is labelled sufficiently with streamline.

7. PLOS authors have the option to publish the peer review history of their article (what does this mean?). If published, this will include your full peer review and any attached files.

Reviewer #1: **Yes: **Pablo Villoslada

Reviewer #2: No

---

## [Author Response · Author response to Decision Letter 1]

9 Dec 2020

Reviewer #1: Authors have addressed properly all my comments and doubts. 

Reviewer #2: The authors have made substantive changes to the manuscript so that their work has become much clearer. I have two issues, which I believe can easily be corrected, with the manuscript as it stands.

Q1. The authors have a whole bunch of metrics which have been shown in their studies, or in others, to correlate with various pathologies. Yet, the authors do not do the correlations themselves with the people in this study. The manuscript itself does an excellent job not overstating what was done. The abstract, on the other hand, does overstate what was done which can be dangerous as the abstract is often the only part of a manuscript that is read by researchers who are looking for some trick to describe their own data. Thus, I suggest changing the overreaching statement “Our result demonstrates close association between an increase of AD and the degree of axonal loss and support the notion that the increase of RD in chronic MS lesions is driven by the combination of two factors: axonal destruction and loss of myelin sheaths in survived axons. While the effect of demyelination dominates in cases of mild axonal loss, neurodegeneration is a primary cause of increased RD in more destructive lesions. These finding highlight the importance of selecting appropriate patient cohorts for clinical trials of pro-remyelinating and neuroprotective therapeutics.” to something more accurate. The summary the authors wrote at the end of the manuscript has a nice statement that could be used instead, “Evaluation of lesions in a sizable cohort of MS patients using the proposed method supports the use of �AD as a marker of axonal loss; and the notion that demyelination and axonal loss independently contribute to the increase of RD in chronic MS lesions.”

A1. Follow reviewer suggestion the last paragraph was removed from Abstract and replaced with following :” Evaluation of lesions in a sizable cohort of MS patients using the proposed method supports the use of �AD as a marker of axonal loss; and the notion that demyelination and axonal loss independently contribute to the increase of RD in chronic MS lesions”.

2. Once again the plots, while much more clear, are not in the standard format for plots. The authors seem to be confusing the title of the plot with what should be on the y-axis. This is true for almost every plot in the figures presented. To make things clear, I will explain what I mean using just the top graph of Figure 2b. There should be no title. The y-axis should be labelled AD (µm2/ms). The x-axis is labelled sufficiently with streamline.

All plots are modified as suggested.

---

## [Decision Letter · Decision Letter 2]

16 Dec 2020

Differentiating axonal loss and demyelination in chronic MS lesions: a novel approach using single streamline diffusivity analysis.

PONE-D-20-27494R2

Dear Dr. Klistorner,

We’re pleased to inform you that your manuscript has been judged scientifically suitable for publication and will be formally accepted for publication once it meets all outstanding technical requirements.

Kind regards,

Fernando de Castro

Academic Editor

PLOS ONE

Additional Editor Comments (optional):

Reviewers' comments:

Reviewer's Responses to Questions

**Comments to the Author**

1. If the authors have adequately addressed your comments raised in a previous round of review and you feel that this manuscript is now acceptable for publication, you may indicate that here to bypass the “Comments to the Author” section, enter your conflict of interest statement in the “Confidential to Editor” section, and submit your "Accept" recommendation.

Reviewer #2: All comments have been addressed

2. Is the manuscript technically sound, and do the data support the conclusions?

Reviewer #2: (No Response)

3. Has the statistical analysis been performed appropriately and rigorously? 

Reviewer #2: (No Response)

4. Have the authors made all data underlying the findings in their manuscript fully available?

Reviewer #2: (No Response)

5. Is the manuscript presented in an intelligible fashion and written in standard English?

Reviewer #2: (No Response)

6. Review Comments to the Author

Reviewer #2: (No Response)

7. PLOS authors have the option to publish the peer review history of their article (what does this mean?). If published, this will include your full peer review and any attached files.

Reviewer #2: No

---

## [Editor Report · Acceptance letter]

21 Dec 2020

PONE-D-20-27494R2 

Differentiating axonal loss and demyelination in chronic MS lesions: a novel approach using single streamline diffusivity analysis. 

Dear Dr. Klistorner:

I'm pleased to inform you that your manuscript has been deemed suitable for publication in PLOS ONE. Congratulations! Your manuscript is now with our production department. 

Kind regards, 

on behalf of

Dr. Fernando de Castro 

Academic Editor

PLOS ONE